**Data Availability Statement:** All relevant data are within the manuscript and its Supporting Information files. In case the journal needs the data

# Survival status and predictors of mortality among low-birthweight neonates admitted to KMC units of five public hospitals in Ethiopia: Frailty survival regression model

**Mesfin Kote Debere** [1,2] *, **Damen Haile Mariam**[1], **Ahmed Ali**[1], **Amha Mekasha**[3], **Grace J. Chan**[4,5]

1 Department of Preventive Medicine, School of Public Health, College of Health Sciences, Addis Ababa University, Addis Ababa, Ethiopia, 2 Department of Epidemiology and Biostatistics, School of Public Health, College of Medicine and Health Sciences, Arba Minch University, Arba Minch, Ethiopia, 3 Department of Pediatrics and Child Health, School of Medicine, College of Health Sciences, Addis Ababa University, Addis Ababa, Ethiopia, 4 Department of Epidemiology, Harvard T.H. Chan School of Public Health, Harvard University, Boston, MA, United States of America, 5 Department of Pediatrics, Boston Children's Hospital, Harvard Medical School, Boston, Massachusetts, United States of Amrica

* messi.kid@gmail.com

## Abstract

### Background

Low birth weight (LBW) and preterm birth are leading causes of under-five and neonatal mortality globally. Data about the timing of death and outcomes for LBW and preterm births are limited in Ethiopia and could be used to strengthen neonatal healthcare. This study describes the incidence of neonatal mortality rates (NMR) stratified by newborn size at birth for gestational age and identifies its predictors at five public hospitals in Ethiopia.

### Methods

A prospective follow-up study enrolled 808 LBW neonates from March 2017 to February 2019. Sex-specific birthweight for gestational age percentile was constructed using Intergrowth 21st charts. Mortality patterns by birthweight for-gestational-age-specific survival curves were compared using the log-rank test and Kaplan-Meier survival curves. A random-effects frailty survival model was employed to identify predictors of time to death.

### Results

Among the 808 newborns, the birthweight distribution was 3.2% <1000 g, 28.3% <1500 g, and 68.1% <2000 g, respectively. Birthweight for gestational age categories were 40.0% both preterm and small for gestational age (SGA), 20.4% term SGA, 35.4% appropriate weight for gestational age, and 4.2% large for gestational age (LGA). The sample included 242 deaths, of which 47.5% were both preterm and SGA. The incidence rate of mortality was 16.17/1000 (95% CI 14.26–18.34) neonatal-days of observation.

Neonatal characteristics independently related to increased risk of time-to-death were male sex (adjusted hazards ratio [AHR] 3.21 95% CI 1.33–7.76), born preterm (AHR 8.56

set, we can upload it, or it can be accessed upon request.

**Funding:** The author(s) received no specific funding for this work.

**Competing interests:** The authors have declared that no competing interests exist.

95% CI 1.59–46.14), having been diagnosed with a complication (AHR 4.68 95% CI 1.49–14.76); some maternal characteristics and newborn care practices (like lack of effective KMC, AHR 3.54 95% CI 1.14–11.02) were also significantly associated with time-to-death.

## Conclusions

High mortality rates were measured for low birthweight neonates–especially those both preterm and SGA births–even in the context of tertiary care. These findings highlight the need for improved quality of neonatal care, especially for the smallest newborns.

## Introduction

The first four weeks after birth is the most vulnerable period for survival, when neonates are exposed to infection and death [1]. Greater than 2.4 million newborn babies died worldwide in 2019 [2]. Around 47% of all under-five mortalities happened in the newborn period, with around 33% dying on the day of birth and almost 75% dying within the first seven days of life [2]. Most of neonatal mortality was in sub-Saharan Africa and southern Asia [3]. The NMR in Ethiopia was 33/1000 live births in 2019 [4] and varied by region from 18 to 48/1,000. Preterm birth (PTB) and asphyxia are the leading causes of neonatal death [2]; 56% of <5 child deaths occur in the neonatal period in Ethiopia [5].

Birthweight is linked with fetal, neonatal, and postnatal mortality; and long-term growth and development [6]. LBW is the principal contributing factor to neonatal mortality [7]. Globally, nearly 15–20% of all births are LBW [8,9], a majority being in resource-poor countries [8]. More than 80% of neonatal deaths in sub-Saharan Africa and South Asia are among small for gestational age (SGA) neonates (65% preterm and 19% term) [10]. LBW has adverse consequences on child survival and may be a substantial risk factor for chronic diseases in future life [11]. The two most central roots of LBW are preterm births (born before 37 completed weeks) and SGA, a proxy for intrauterine growth restriction [12–14]. According to the World Health Organization (WHO), SGA is demarcated as a birthweight under the 10[th] percentile for gestational age via gender-category-specific reference population [6,15]. Also, obtaining accurate gestational age data is often challenging in places like Ethiopia [16].

Even if both SGA and preterm births have mutual risks, the relative significance of those risk factors is different [17]. Merging preterm birth and SGA as LBW may inhibit the advancement of protective mechanisms [17]. A limited number of studies have scrutinized the mortality risk among preterm births and SGA neonates of LBW [7] and the risk and timing of death have not been as well studied in developing countries like Ethiopia [18]. Mortality rates stratified by birth size-for-gestational age are also important in assessing perinatal services and in counseling parents [19]. However, there are little data on mortality rates stratified by birth size-for-gestational-age and independent predictors of time-to-death in these low birthweight (<2000 grams) babies in Ethiopia. Therefore, this study aimed to determine the incidence of neonatal mortality rates and identify its independent predictors among babies born <2000 grams born or treated at five public hospitals in Ethiopia.

## Methods and materials

### Study design, setting, and population

A prospective follow-up study was conducted among a cohort of babies born <2000g that could have been born or brought there because of some complication to the five study

hospitals in the Oromia Region and Addis Ababa City from March 2017 to February 2019. The health facilities included were Assela Teaching and Referral Hospital (158 babies), Kersa Primary Hospital (22 babies), Bekoji Primary Hospital (26 babies), Batu General Hospital (236 babies) from Oromia Region, and Tirunesh-Beijing General Hospital (366 babies) from Addis Ababa City. Births that took place within private hospitals/clinics and health centers in the study catchment area were also included.

All mother-newborn pairs were followed independently after birth until four weeks of life or death or transferred out to a facility outside of the study area after three days of life, whichever occurred first. The exclusion criteria were: neonates whose birthweight was not measured within 72 h of life; newborns whose gestational age could not be established either by last menstrual period or first-trimester ultrasound; and newborns who were referred to a facility out of the study area within three days of life.

## Sample size determination and sampling procedure

However, we included everyone during the study period (n = 808) using cluster sampling, and the minimum required sample size of **752** was calculated using one group dichotomous outcome variable (mortality: died, censored) using the STATA statistical package. Assumptions used for the calculation were NMR of 28% based on previous estimates($P_o$) [20]; setting the rate of deaths of neonates in the area as high ($P_1$ = 34%) to make the sample size large enough to detect the true incidence; 95% confidence interval, $\alpha$ = 0.05, and 80% power; about 10% lost-to-follow-up; and intra-cluster correlation (1.5). The adequacy of the sample size computed to determine the incidence rate of mortality was also checked whether it was sufficient or not to identify predictors of mortality. Using a two-sample comparison of survivor functions (Log-rank test, Freedman method), the number of events (E) that could be needed for the study was calculated using: $E = ((Z_{\alpha/2} + Z_\beta)^2)/((\ln(HR))^2\, pq)$. Using Freedman principles, survival probability (p) is equal to the rate of an event (q) at the end of the study period (the median survival = 50%), i.e. p = q. To calculate the total number of events (E) that could be needed to compare the two groups (exposed and unexposed/1:1 ratio), we take $\alpha$ = 0.05 and $\beta$ = 0.20. With these values of $\alpha$ and $\beta$, $Z_{\alpha/2}$ = 1.96 and $Z_\beta$ = 0.84, and taking the HR 1.65 from a previous study [20], and 10% withdrawal, the number of events (E) becomes 132. Then, the total number of participants needed (n) for the survival study was calculated using n = E/Pr (event), and it becomes 407 by considering the clustering effect of 1.5. Thus, the sample size calculated for estimating the incidence rate of mortality was sufficiently enough to identify independent predictors of time-to-death.

## Outcome variable

The outcome variable was time-to-death. Event indicators were defined as 1 = died and 0 = censored.

## Exposure variables

The following exposure variables were considered:

- Distal (background) factors- marital status, mother's occupation, mother's education, father's occupation, father's education, father's age, income, and the number of children.

- Proximate determinants:

  ○ Maternal factors (age, parity, history of abortion and stillbirth, maternal complications);

○ Neonatal factors (sex of the newborn, birth type, birth interval, neonatal complications, sickness status, gestational age at birth, birthweight, birth size-for-gestational-age (categorized as SGA (<10th percentile), AGA (10th percentile to 90th percentile) or LGA (>90th percentile) using the sex-and age-specific Intergrowth-21st birthweight charts) [21].

○ Newborn feeding practices (hours after birth first put the baby to the breast, breastfeeding in the last 24 hours during KMC initiation).

○ Delivery factors (place of birth, mode of delivery, who assisted the delivery.

○ Neonatal care practices- KMC, duration of skin-to-skin contact (SSC), wash after birth). The details of the adapted framework are shown in **S1 Fig**.

## Operational definitions

Early neonatal mortality rate (ENMR) was described as the probability of death occurring in the first seven days of life. Late neonatal mortality rate (LNMR), the chance of death between 8 and 28 completed days [22].

Maternal complication is categorized into 'Yes = 1' or 'No = 0', which is considered present if the mother had an obstetric hemorrhage, puerperal sepsis, and pyrexia, prolonged labor, pre-eclampsia-eclampsia, malpresentation and malposition, premature rupture of membrane (PROM), obstructed labor, and retained placenta.

Neonatal complication is categorized into 'Yes = 1' or 'No = 0', which is considered present if the neonate had asphyxia, infection, hypothermia, jaundice, feeding difficulties, and other rare complications during birth.

Kangaroo mother care (KMC) involves the provision of skin-to-skin care (SSC) by the mother or caregiver for as long as possible during day and night, along with exclusive breast-feeding (EBF) or breastmilk feeding. A minimum of eight hours of SSC along with EBF or breastmilk feeding over the previous 24 hours was required for it to be categorized as effective KMC [23].

## Data collection instruments and process

Data were collected using pre-tested interviewer-administered questionnaires and data abstraction tools, which were adapted from the literature [23,24]. Data were derived from hospital records, part from the daily or periodic observation, and part from interviews with the mothers/caregivers. Birthweight was extracted from the delivery register. On admission, for every live birth LBW, neonate data were collected. Neonatal survival was also monitored during the subsequent follow-up visits after discharge (post-facility) until 28 days of life if discharged before 28 days of life. If the baby was discharged before 28 days of life, there were home visits at seven days of life, seven days post-discharge, and 29 days of life to track information. For newborns who survived for more than 28 days, their follow-up time was censored at the end of their 28 days of life. Twins and triplets were treated as independent observations.

## Data management and statistical analysis

REDCap application system was used to collect the data. STATA version 14 was used for cleaning and analyzing the data (S1 Dataset). Descriptive statistics were computed. Sex-specific birthweight percentiles or z-scores were generated using Intergrowth 21st charts [21]. The NMR, ENMR, and LNMR rates were determined by taking the total number of live births weighing < 2000g at birth as a denominator and reported as per 100 live births. The overall

and birth size-for-gestational-age-specific incidence rates of mortality were calculated using person-time methods. Life tables were constructed to estimate the probabilities of survival along with Kaplan-Meier survival curves. A log-rank test was constructed to compare the survivor functions. In this study, some covariates were not measured for various reasons. To account for these unobserved covariates, a random-effects survival model, parametric individual/univariate frailty regression model, was employed to identify independent predictors of time-to-death. Here, frailty describes variations that were not explained by observed covariates. A one-parameter gamma frailty distribution with mean 1 and variance θ, and Weibull distribution for the baseline hazard, was employed.

A bivariate parametric individual frailty regression model was fitted first and those independent variables that became significant on the bivariate regression at 25% level of significance were carried forward to multivariable regression. Backward stepwise regression was used to select variables. A 95% CI and 5% level of significance were used. The frailty term (α) and the two forms of shape parameter (p and log (p) were reported and interpreted for the final model. Model assumptions were checked by a graphic approach. The appropriateness of a Weibull individual frailty regression model was determined by plotting: ln [-lnS(t)] vs ln(t). To select the best-fitting model (post estimation), the information criterion, Akaike and Bayesian (AIC and BIC) was used along with the Log-likelihood test.

The occurrence of multicollinearity between variables was evaluated using variance inflation factors (VIF>10 taken as the presence of collinearity). None of the predictor variables were collinear.

## Ethical considerations

Ethical approval was obtained from the institutional review boards of the College of Health Sciences, Addis Ababa University, and Harvard T.H. Chan School of Public Health. Permissions were obtained from the Addis Ababa-City-Administration-Health-Bureau and Oromia Regional Health Bureau and hospital administrations. Written informed consent was obtained from mother/caregivers of neonates in the study hospitals. The participants were assured of confidentiality as they participated in the investigation.

## Results

### Socio-demographic characteristics of participants

A total of 1092 mother-newborn pairs were enrolled in this study. 11.72% (n = 128) were excluded because of missing gestational age at birth. Additionally, 14.28% (n = 156) were excluded since referred to a facility out of the study area within three days of life. A total of 808 mother-newborn pairs, of whom 436 were admitted to the NICU, participated in this study (**Fig 1**). There were no differences between babies who were included in this analysis and who were excluded from this analysis for their basic characteristics like sex of the baby ($X^2$ = 0.0450, P-value = 0.832), birth size-for-gestational-age ($X^2$ = 1.01 P-value = 0.61) and mother's age at birth ($X^2$ = 47.27 P-value = 0.14). Baseline characteristics are shown in Table 1. Almost all 743 (94%) of the mothers were married, while only 40 (4.95%) were single and 7 (0.87%) were divorced/widowed (Table 1). About 205 (25.37%) of the mothers had no education, and 313 (39.74%) of the mothers had completed primary school (grades 1 to 8). Closely one-third (30.57%) of the mothers had completed secondary school (grade 9 to 12) while 43 (5.32%) were above secondary. The majority of the mothers, 520 (64.36%) were housewives by occupation followed by farmers, 68 (8.42%). The newborns were followed for a minimum of 1 day and a maximum of 28 days. The mean time-to-death for the whole cohort was 18.52 days

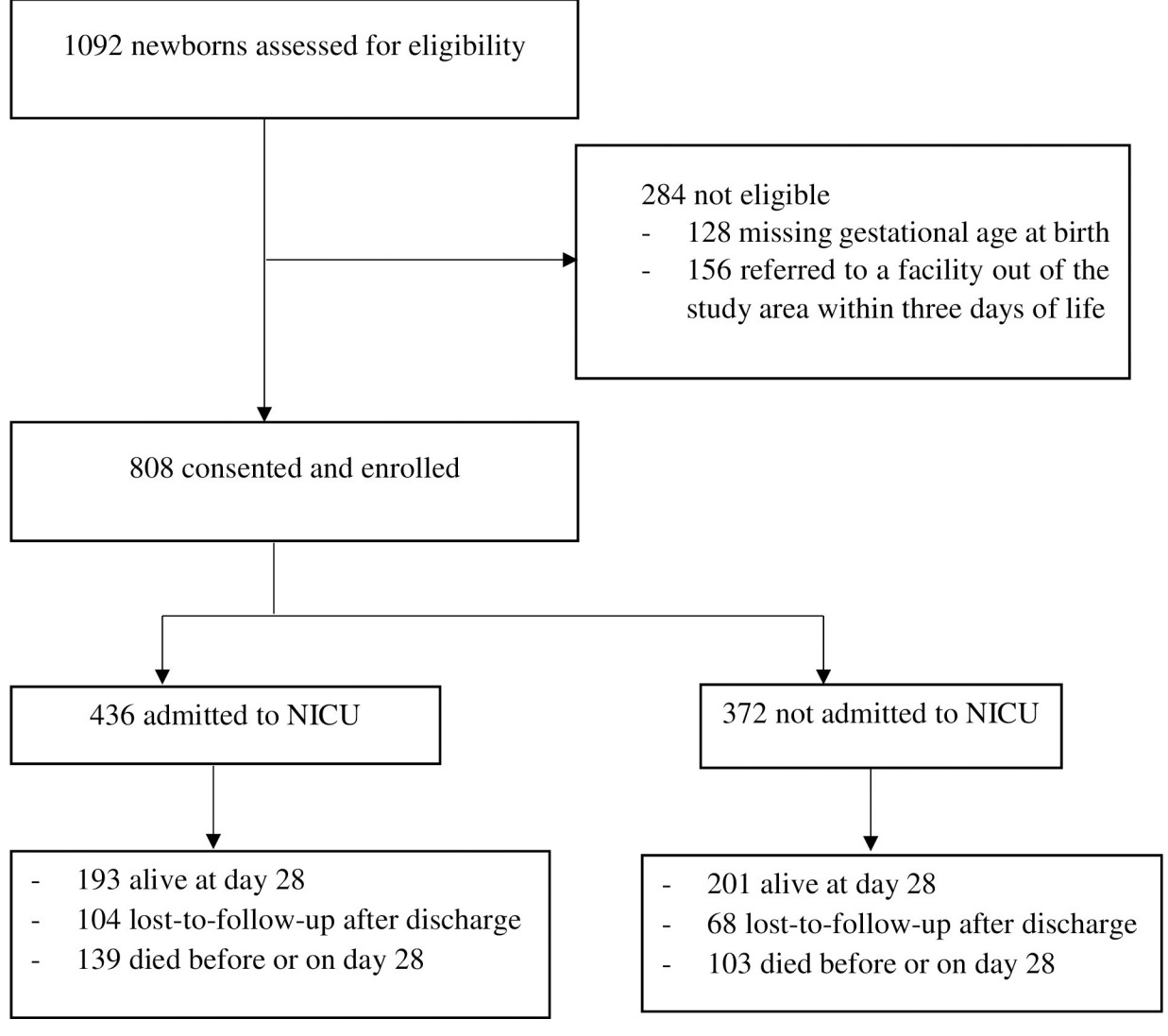

**Fig 1. Study flow diagram of the overall study process.**

(SD = 10.65). The 808 newborns add up to a total time of 14967 neonatal days (NDs) of observation.

## Maternal characteristics

Around 45% of the mothers were primigravida and 42% and nearly 13% of the mothers were multi-gravida, (gravida II to IV and above), respectively (Table 2). Almost half (51.24%) of the newborns were from primiparous mothers. About 138 (17.00%) mothers had a previous history of abortion, and 4.83% of the mothers had a previous history of stillbirth. Just 7.18%, 30.94%, 35.40%, 18.81%, and 7.67% of the LBW live birth babies were from teenage mothers (age less than 20 years), 20 to 24 years, 25 to 29 years, 30 to 35 years, and more than 35 years, respectively. Nearly 70% of the mothers gave birth at government hospitals (all types), whereas 6.30% gave birth at home, 22.28% at health centers/health posts, and 1.49% at private hospitals/clinics.

**Table 1. Sociodemographic characteristics of mothers and fathers whose babies were admitted to KMC units of study hospitals in Oromia Regional State and AA City, Ethiopia, 2017–2019.**

| Characteristics | Response categories | Number (%) | Time at risk | Incidence rate/$10^3$ |
|---|---|---|---|---|
| Mother's marital status | Single | 40 (4.95) | 734 | 19.07 |
| | Married | 761 (94.18) | 14109 | 15.81 |
| | Divorced/widowed | 7 (0.87) | 124 | 40.32 |
| Mother's education | No education | 205 (25.37) | 3477 | 24.73 |
| | Primary (grades 1–8) | 313 (38.74) | 5821 | 14.95 |
| | Secondary (grades 9–12) | 247 (30.57) | 4861 | 11.73 |
| | Above secondary | 43 (5.32) | 808 | 14.85 |
| Mother's occupation | Agriculture | 68 (8.42) | 1100 | 22.73 |
| | Housewife | 520 (64.36) | 9670 | 16.03 |
| | Professional work | 53 (6.56) | 1021 | 12.73 |
| | Sales and services | 48 (5.94) | 1053 | 10.45 |
| | Skilled manual | 53 (6.56) | 963 | 14.54 |
| | Unemployed | 16 (1.98) | 215 | 41.86 |
| | Unskilled manual | 50 (6.19) | 945 | 15.87 |
| Father's age (years) | < 25 | 62 (7.67) | 1110 | 16.22 |
| | 25 to 29 | 218 (26.98) | 3905 | 17.67 |
| | 30 to 34 | 221 (27.35) | 4610 | 9.76 |
| | ≥ 35 | 278 (34.41) | 4957 | 18.56 |
| | Missing | 29 (3.59) | 385 | 46.73 |
| Father's education | No Education | 106 (13.12) | 1880 | 20.21 |
| | Primary (grade 1–8) | 276 (34.16) | 4884 | 17.81 |
| | Secondary (grade 9–12) | 322 (39.85) | 6156 | 14.29 |
| | Above secondary | 75 (9.28) | 1588 | 8.82 |
| | Missing | 29 (3.59) | 459 | 32.68 |
| Number of living children | < 2 | 306 (37.87) | 6112 | 10.96 |
| | 2 to 4 | 386 (47.77) | 7222 | 15.09 |
| | 5+ | 116 (14.36) | 1633 | 40.42 |
| Av. monthly income (n = 708) | Mean = 3200.01, SD = 4471.03 | | 13138 | 16.06 |

## Newborn-related characteristics and newborn care and feeding practices

About 54% of newborns were males and the remaining 46% were females, with a male-to-female ratio of 1.18:1 (Table 3). 3.09% of the newborn babies included in this study were extremely low birth weight (ELBW) while 28.59% and one-third (68.32%) of the babies were very low birth weight (VLBW) and LBW (1500 to 1999 g), respectively. 15.59% of the babies were term while 84.41% were preterm. Of the 808 newborn babies, 488 (60.40%) of them were SGA, 286 (35.40%) were AGA and the remaining 34 (4.21%) were LGA babies. 323 (39.98%) newborns were born both preterm and SGA. About 5,435 NDs of observation were for pre-term-SGA babies, 3142 NDs for term-SGA, 5654 NDs for AGA, and 736 NDs of observation for LGA babies.

When we look at the stratified neonatal complications (by sex of the newborns), almost in all of the complications the highest frequency was observed in male neonates (Fig 2). The average hours of SSC given to the newborns per day during KMC initiation were 9.96 hours (SD = 4.48) while it was 11.49 hours (SD = 4.69) immediately before discharge. About 34.90% (319/808) and 14.73% (119/808) of the newborns had received greater or equal to 8 hours and less than 8 hours of SSC per day during KMC initiation, respectively. About 45.30% (366/808)

**Table 2. Maternal obstetric characteristics of mothers of newborns admitted to KMC facilities of five public hospitals in Oromia Regional State and AA City, Ethiopia, 20017–2019, (N = 808).**

| Characteristics | Response categories | Number (%) | Died | Time at risk | Rate/10³ |
|---|---|---|---|---|---|
| Gravidity | Primi-gravida | 369 (45.67) | 106 | 6901 | 15.36 |
| | Multi-gravida (II to IV) | 338 (41.83) | 99 | 6302 | 15.71 |
| | Grand multi-gravida V+ | 101 (12.50) | 37 | 1764 | 20.98 |
| Parity | Primiparous | 414 (51.24) | 115 | 7748 | 14.84 |
| | Multi-parous (2–4) | 316 (39.11) | 94 | 5906 | 15.92 |
| | Grand multi-parous (5+) | 78 (9.65) | 33 | 1313 | 25.13 |
| History of abortion | No | 670 (82.92) | 164 | 13151 | 12.47 |
| | Yes | 138 (17.00) | 78 | 1816 | 42.95 |
| Previous history of stillbirth | No | 755 (93.44) | 222 | 14021 | 15.83 |
| | Yes | 39 (4.83) | 13 | 711 | 18.28 |
| | Missing | 14 (1.73) | 7 | 235 | 29.79 |
| Mother's age at birth | < 20 years | 58 (7.18) | 21 | 992 | 21.17 |
| | 20 to 24 years | 250 (30.94) | 77 | 4505 | 17.09 |
| | 25 to 29 Years | 286 (35.40) | 83 | 5534 | 14.99 |
| | 30 to 34 Years | 152 (18.81) | 45 | 2741 | 16.42 |
| | 35+ Years | 62 (7.67) | 16 | 1195 | 13.39 |
| Place of birth | Government hospitals all type | 547 (67.70) | 132 | 10736 | 12.29 |
| | Home | 51 (6.30) | 39 | 554 | 70.39 |
| | Health center/ health post | 180 (22.28) | 65 | 3090 | 21.04 |
| | Other | 12 (1.49) | 3 | 208 | 14.42 |
| | Missing | 18 (2.23) | 3 | 379 | 7.92 |
| Mode of delivery | Spontaneous vaginal delivery | 685 (84.78) | 204 | 12738 | 16.02 |
| | Assisted vaginal delivery | 33 (4.08) | 10 | 615 | 16.26 |
| | Cesarean section | 88 (10.89) | 28 | 1578 | 17.74 |
| | Missing | 2 (0.25) | 0 | | |
| Who conducted the delivery | Doctor or health officer | 146 (18.07) | 57 | 2424 | 23.52 |
| | Nurse or midwife | 604 (74.75) | 164 | 11463 | 14.31 |
| | Traditional birth attendant | 8 (0.99) | 2 | 182 | 10.99 |
| | Community health worker | 13 (1.61) | 4 | 253 | 15.81 |
| | Relative and neighbor | 12 (1.49) | 5 | 215 | 23.26 |
| | Mother herself | 20 (2.48) | 8 | 365 | 21.92 |
| | Missing | 5 (0.62) | 2 | 65 | 30.77 |
| Maternal complications | No | 673 (83.29) | 193 | 12598 | 15.32 |
| | Yes | 135 (16.71) | 49 | 2369 | 20.68 |
| Types of maternal complications | Eclampsia | 114 (14.11) | 38 | 2030 | 18.72 |
| | Antepartum hemorrhage | 9 (1.11) | 3 | 162 | 18.52 |
| | Postpartum hemorrhage | 6 (0.74) | 6 | 39 | 15.39 |
| | Others* | 14 (1.73) | 4 | 276 | 14.49 |

(*Includes: PROM (3), retained placenta (1), mal-presentation (2), HIV/AIDS (3), anemia (5).

and 10.52% (85/808) of the newborns had received greater or equal to 8 hours and less than 8 hours of SSC per day before discharge from the facility, respectively. Just 68.81% (556/808) of the newborns initiated breastfeeding while 29.83% (241/808) did not initiate within 24 hours of birth. About 24.88% (201/808), 29.08% (235/808), and 14.85% (120/808) of the newborns were put into the breast after birth for the first time at less than one hour, between 1 to 24

**Table 3. Characteristics of low-birth-weight neonates admitted to KMC facilities of five public hospitals in Oromia Regional State and Addis Ababa City, Ethiopia, 20017–2019, (n = 808).**

| Characteristics | Response categories | Number (%) | Died | Time at risk | Incidence rate/$10^3$ |
|---|---|---|---|---|---|
| Newborn's sex | Male | 437 (54.08) | 152 | 7780 | 19.54 |
| | Female | 371 (45.92) | 90 | 7187 | 12.52 |
| Birthweight group (g) | ELBW (< 1000 g) | 25 (3.09) | 22 | 217 | 101.38 |
| | VLBW (1000 g—1499 g) | 231 (28.59) | 115 | 3448 | 33.35 |
| | LBW (1500 g—1999 g) | 552 (68.32) | 105 | 11302 | 9.29 |
| Gestational age at birth | Term | 165 (20.42) | 61 | 2839 | 21.49 |
| | Preterm | 643 (79.58) | 181 | 12128 | 14.92 |
| Birth size-for-gestational-age | Preterm-SGA | 323 (39.98) | 115 | 5435 | 21.16 |
| | Term-SGA | 165 (20.42) | 61 | 3142 | 19.41 |
| | AGA | 286 (35.40) | 61 | 5654 | 10.79 |
| | LGA | 34 (4.21) | 5 | 736 | 6.79 |
| Birth order | 1st born | 431 (53.34) | 124 | 8015 | 15.47 |
| | 2nd born | 157 (19.43) | 44 | 2862 | 15.37 |
| | 3rd to 4th born | 143 (17.70) | 40 | 2781 | 14.38 |
| | 5th+ born | 77 (9.53) | 34 | 1309 | 25.97 |
| Birth interval (years) | 1st born | 431 (53.34) | 124 | 8015 | 15.47 |
| | < 3 Years | 149 (18.44) | 56 | 2754 | 20.33 |
| | 3+ Years | 224 (27.72) | 62 | 4120 | 15.05 |
| | Missing | 4 (1.06) | 0 | 78 | - |
| Multiple births | Singleton | 531 (65.72) | 165 | 9645 | 17.11 |
| | Twins | 260 (32.18) | 72 | 5032 | 14.31 |
| | Triplets | 16 (1.98) | 5 | 262 | 19.08 |
| | Missing | 1 (0.12) | 0 | 28 | |
| Length of hospital stay | < = 3 days | 95 (11.76) | 68 | 865 | 78.61 |
| | 4 to 7 days | 181 (22.40) | 75 | 3377 | 22.21 |
| | > 7 days | 434 (53.71) | 39 | 9622 | 4.05 |
| | Missing | 98 (12.13) | 60 | 1103 | 54.39 |
| Neonatal complications | Yes | 391 (48.39) | 155 | 6489 | 23.89 |
| | No | 201 (24.88) | 17 | 4750 | 3.578 |
| | Missing | 216 (26.73) | 70 | 3728 | 18.78 |
| Wash immediately after birth | Yes | 187 (23.14) | 128 | 2255 | 56.76 |
| | No | 330 (40.84) | 34 | 7653 | 4.44 |
| | Missing | 291 (36.01) | 80 | 5059 | 15.81 |
| Hours after birth, first put the baby on the breast | BF has not yet initiated | 239 (29.58) | 150 | 2849 | 52.65 |
| | BF initiated at <1hr | 200 (24.75) | 27 | 4502 | 5.99 |
| | Initiated within 1 to < = 24 hrs | 232 (28.71) | 21 | 5226 | 4.02 |
| | BF was initiated after 24 hours | 119 (14.73) | 41 | 2011 | 20.39 |
| | Missing | 18 (2.23) | 3 | 379 | 7.92 |

hours and greater than 24 hours of duration, respectively while 1.36% (11/808) were missing information. Around 65.10% (526/808) of the newborns had received colostrum. The coverage of effective KMC (at least 8 hours of SSC and breastfeeding in the last 24 hours) during KMC initiation was 34.90% (282/808) while it was 44.93% (363/808) immediately before discharge.

Six hundred four (74.75%) of the newborns were sick. About 470 (58.17%) of the newborns had received oxygen therapy, 327 (40.47%) received antibiotics, 238 (29.46%) received glucose and 436 (53.96%) of the newborns had been placed in a radiant warmer/incubator. Four

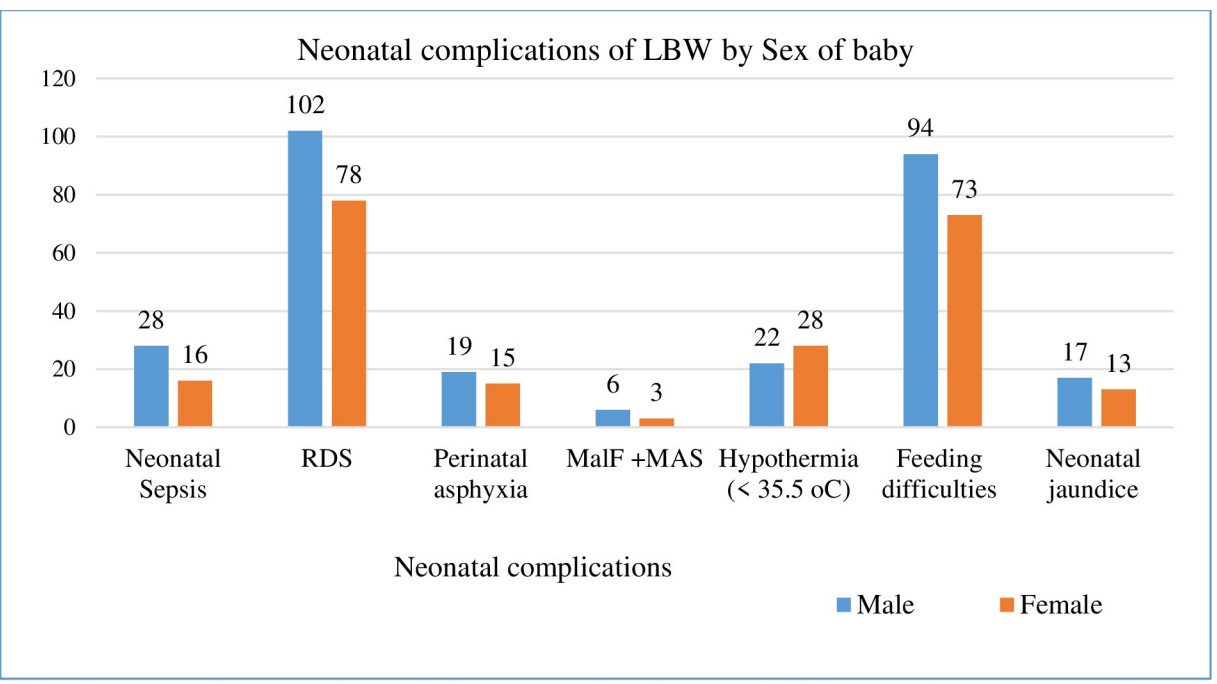

RDS= Respiratory distress syndrome, MalF = congenital malformation, MAS = meconium aspiration syndrome

**Fig 2. Frequency distribution of neonatal complications stratified by sex of the newborns among LBW neonates admitted to KMC units of five selected public hospitals in the Oromia Region and Addis Ababa City, Ethiopia, 20017–2019.**

(0.50%) of the babies were given breastmilk from another mother whereas 54 (6.68%) of the newborns had received any milk other than breastmilk such as powdered, or fresh animal milk. Sixty-two (7.67%) of the babies were given other fluids (like juice, tea, sugar, water, and honey). There was a statistical significance difference between newborns' sickness status and breastfeeding initiation (chi-square test = 116.13, p-value = 0.001). Out of 604 sick newborn babies, 226 (37.42%) of them did not initiate breastfeeding while 280 (46.36%) of them initiated early (within 24 hours of birth) and 90 (14.90%) lately (after 24 hours of birth).

### Neonatal survival status or mortality rates

The overall incidence rate of mortality was 16.17/1000 NDs of observation (95% CI 14.26–18.34). SGA babies were at risk to die within a short time when compared with AGA and LGA babies. The rate of mortality for preterm-SGA newborns was 21.16/1000 NDs (95% CI 17.49–25.34) while it was 19.41/1000 NDs (95% CI 14.88–24.87), 10.79/1000 NDs (95% CI 8.26–13.83), and 6.79/1000 NDs (95% CI 2.21–15.78) for term-SGA, AGA, and LGA newborns, respectively and the difference was statistically significant with a p-value of 0.001 (S2 Fig). The cumulative failure of babies born <2000g was low in the first day of life, which rises as follow-up time upturns. The percentage of failure was 4.83%, 13.49%, 23.28%, and 33.53% at the end of the first day of life, three days of life, 7 days of life, and 28 days of life, respectively (S2 Fig). During the study period, there were a total of 242 (29.95%) neonatal deaths which falls within a 95% CI of 26.81 to 33.24% corresponding to NMR of 299.50 per 1000 live births. The NMR was 22.40% for preterm LBW and 7.55% for term LBW babies. The death rate in the first 3 days of life was 13.49% (95% CI 11.20–16.04) (Table 4). The ENMR and LNMR were 23.02% (95% CI 20.16–26.01) and 6.68% (95% CI 5.06–8.63), respectively. The NMRs were significantly different among the three categories of newborn birth size-for-gestational-age. Out of

**Table 4. Age-specific mortality rates stratified by newborn size among LBW neonates admitted to KMC units of five public hospitals in Oromia Region and Addis Ababa City, 2017–2019.**

| Age-specific mortality rates[*] | Stratified by newborn size by gestational age | | | | | Total deaths, n (%) |
|---|---|---|---|---|---|---|
| | Preterm SGA, n (%) | Term SGA, n (%) | AGA, n (%) | LGA, n (%) | P-value | |
| Death in the first 3 days of life | 47 (14.55) | 37 (22.42) | 24 (8.39) | 1 (2.94) | 0.002 | **109 (13.49)** |
| ENMRs (deaths ≤ 7 days) | 88 (27.24) | 54 (32.73) | 43 (15.03) | 3 (8.82) | 0.001 | **186 (23.02)** |
| LNMRs (8–28 days of life) | 27 (8.36) | 7 (4.24) | 18 (6.29) | 2 (5.88) | 0.92 | **54 (6.68)** |
| NMR (death in the 28 days of life) | **115 (35.60)** | **61 (36.97)** | **61 (7.55)** | **5 (0.62)** | 0.001 | **242 (29.95)** |

[*]These rates were cumulative incidences, considering the total number of newborns enrolled in each stratum as the denominator.

242 deaths of newborns, 181 (74.79%) and 176 (72.73%) of them were born preterm and SGA, respectively while 115 (47.52%) of them were born both preterm and SGA.

**Fig 3** illustrates the cumulative probability of surviving at the end of different time points. The overall cumulative probability of surviving at the end of 3, 6, and 21 days of life were 89.73% (95% CI 87.42–91.63), 80.19% (95% CI 77.27–82.78), and 71.95% (95% CI 68.67–74.96), respectively. The cumulative probability of surviving at the end of the follow-up time

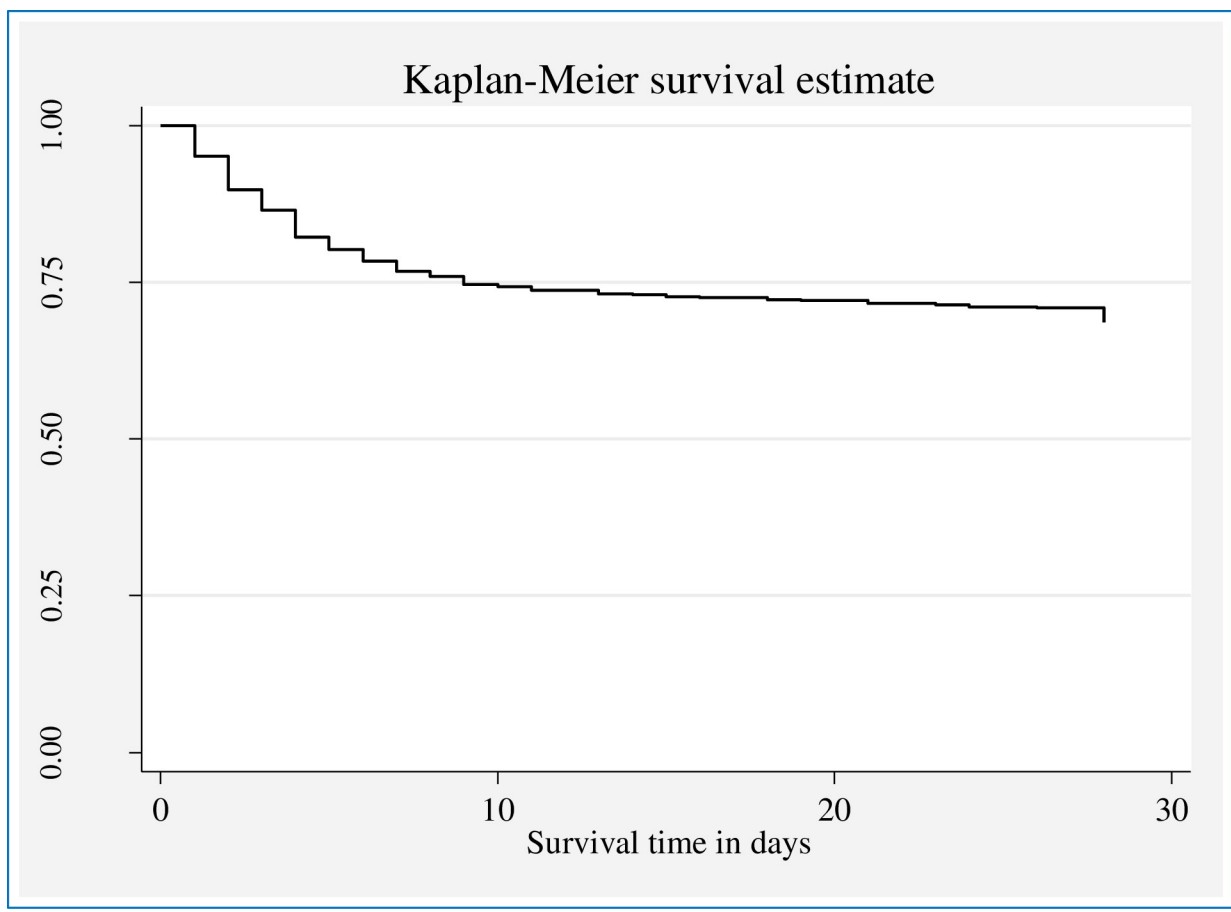

**Fig 3. Survival curve showing the overall cumulative probability of surviving among LBW neonates admitted to KMC units of five Public Hospitals in the Oromia Region and Addis Ababa City Administration, Ethiopia, 2017–2019.**

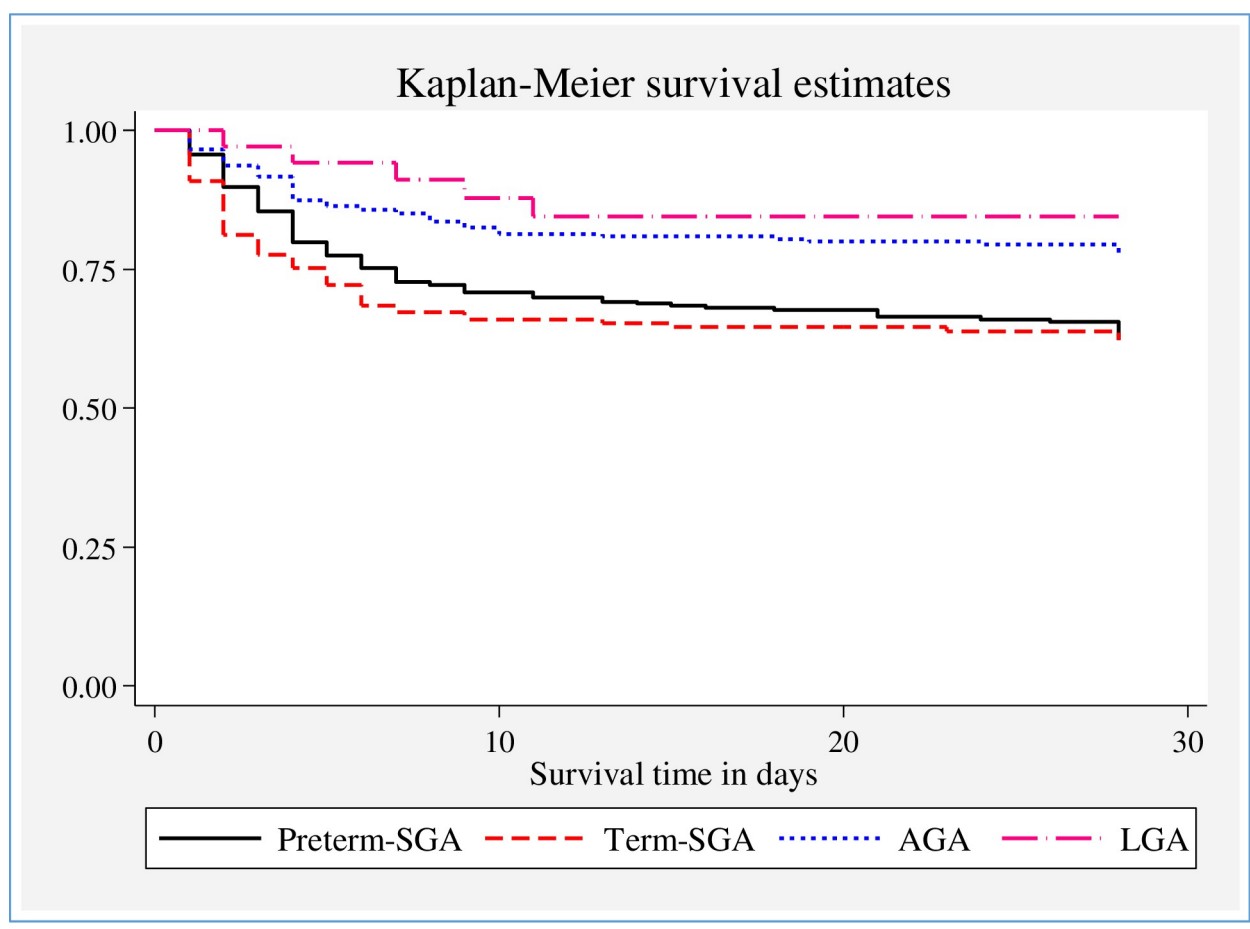

**Fig 4. Survival curve showing the cumulative probability of surviving stratified by newborn size-for-gestational-age among LBW neonates admitted to KMC units of five public hospitals in Oromia Region and Addis Ababa City, Ethiopia, 2017–2019).**

(28 days of life) was 66.41%; 95% CI 62.47–70.04%. In the first 6 days of life, the curve drives like a steep slope and increasingly down which signifies high neonatal death and low probability of survival in this specified period. Whereas, over the next eight days (7 to 14), the probability of surviving has marginally enhanced and the curve fell progressively. After 15 days of life, the curve fell more gently which designates the survival rate of the neonates increased; while after 21 days of life the curve persisted constantly which implies nearly no deaths (**Fig 3**).

The cumulative probability of surviving was also computed separately for the categories of birth size-for-gestational-age. The SGA categories of the newborns had a lower cumulative probability of surviving throughout the neonatal period compared with both AGA and LGA babies The cumulative probability of surviving along with their corresponding 95% CI at the end of 3, 6, 21 and 28 days of life for preterm-SGA babies were 89.74% (95% CI 85.87–92.59), 76.80% (95% CI 71.77–81.06, 66.60% (95% CI 61.03–71.57and 58.41% (95% CI 51.48–64.70), respectively (S1 Table). As the curve in Fig 4 demonstrates, there was a statistically significant variation in survival among term-SGA vs preterm-SGA; and AGA vs LGA categories. The curves for preterm-SGA vs AGA and term-SGA vs LGA in the graph did not coincide with each other which demonstrates there was a disparity in survival among the groups. The cumulative probability of surviving was significantly higher for LGA newborns compared with SGA and AGA over time. But there was no statistically significant variation in survival among term-SGA and preterm-SGA (**Fig 4**).

**Table 5. Log-rank test for survival difference for neonatal mortality between covariates among LBW neonates admitted to KMC units of five Public Hospitals in the Oromia Region and Addis Ababa City, Ethiopia, 2017–2019.**

| Comparison group | Categories | Events observed | Events expected | Log rank test | |
|---|---|---|---|---|---|
| | | | | $X^2$ | P-value |
| Sex of the newborn | Male | 152 | 126.89 | 10.83 | 0.001 |
| | Female | 90 | 115.11 | | |
| Birthweight | ELBW (< 1000 g) | 22 | 5.55 | 131.98 | 0.001 |
| | VLBW (1000 to < 1499 g) | 115 | 60.57 | | |
| | LBW (1500 to < 2000 g) | 105 | 175.88 | | |
| Birthweight-for-gestational-age | Preterm-SGA | 115 | 94.75 | 22.82 | 0.001 |
| | Term-SGA | 61 | 46.04 | | |
| | AGA | 61 | 89.94 | | |
| | LGA | 5 | 11.28 | | |
| Neonatal complications | Yes | 155 | 105.06 | 63.51 | 0.001 |
| | No | 17 | 66.94 | | |
| Effective KMC | Yes | 24 | 121.40 | 166.61 | 0.001 |
| | No | 215 | 117.60 | | |
| Hours after birth, first put the baby on the breast | BF has not yet initiated | 150 | 57.13 | 231.70 | 0.001 |
| | Initiated <1 hr | 27 | 68.26 | | |
| | Initiated within 1 to < = 24 hrs | 21 | 79.23 | | |
| | BF was initiated after 24 hrs | 41 | 34.38 | | |

## Mortality and comparison of time to death for different categorical variables

The presence of significant differences in mortality experiences between LBW neonates of males and females as well as the different categories of covariates were performed using the Log-rank test. There was an overall statistically significant difference between males and females ($X^2$ = 10.83, p-value = 0.001), and among the different categories of birthweight, birth size-for-gestational-age, neonatal complications, effective KMC, and hours after birth first put the baby to the breast (Table 5). Males experienced higher mortality than females. Neonates having ELBW and VLBW had a higher risk of death compared with 1500-<2000g. Out of 25, 22 of the ELBW babies died (880/1000 livebirths of ELBW). The mortality rate significantly declined for VLBW babies (497.84/1000 live births) and LBW babies (190.22/1000 live births) compared with ELBW babies. ELBW babies accounted for 9.09% (22/242) of the deceases while almost half, 47.52% (115/242) of the deceases were accounted by VLBW babies. The median hazard times-to-death of ELBW and VLBW neonates were shorter than the time-to-death of neonates having LBW (S3 Fig).

As indicated in Table 6 below, overall, there was a significant difference in mortality experiences between the different categories of birth size-for-gestational-age ($X^2$ = 22.82, p-value = 0.001). Newborns who had neonatal complications at birth were at a higher risk to die within a short time compared with those who hadn't complications. The incidence rate of mortality was 23.76 /$10^3$ and the difference was significant with a p-value of 0.001. Likewise, newborns who had got effective KMC were more likely to die within a short period of time compared with those who hadn't. The median hazard time to death for neonates who hadn't got effective KMC was shorter than those who had effective KMC (p-value = 0.001). The incidence rate of mortality was 3.03/$10^3$ for neonates who had effective KMC while it was 32.71/$10^3$ for neonates who hadn't got effective KMC. Moreover, newborns who had not yet initiated BF were more

**Table 6.** Multivariable individual frailty regression model showing the crude hazard ratios (CHRs) and AHRs in the study of predictors of mortality among low-birth-weight neonates admitted to KMC units of five public hospitals in Oromia Region and Addis Ababa City, Ethiopia, 2017–2019.

| Characteristics | Neonatal mortality | | CHR (95% CI) | AHR (95% CI) |
|---|---|---|---|---|
| | Censored (n = 566) n(%) | Died (n = 242) n(%) | | |
| **Gravidity** | | | | |
| Primigravida | 263 (71.27) | 106 (28.73) | 1.31 (0.04, 38.54 | 1.18 (0.48, 2.90) |
| Grand multi-gravida(V+) | 64 (63.37) | 37 (36.63) | 0.25 (0.01, 6.22) | 0.55 (0.15, 2.03) |
| Multi-gravida (II-IV) (ref). | 239 (70.71) | 99 (29.29) | 1.00 | 1.00 |
| **Mother's age at birth (n = 790)** | | | | |
| < 25 years | 204 (67.77) | 97 (32.23) | 3.27 (0.04, 25.20 | 1.38 (0.46, 4.11) |
| 25 to 29 years | 199 (70.82) | 82 (29.18) | 3.96 (0.09, 17.38) | 0.80 (0.30, 2.19) |
| > = 35 years | 45 (73.77) | 16 (26.23) | 2.91 (0.09, 8.60) | 0.27 (0.05, 1.43) |
| 30 to 34 years (ref) | 103 (70.07) | 44 (29.93) | 1.00 | 1.00 |
| **Number of living children** | | | | |
| 2 to 4 | 277 (71.76) | 109 (28.24) | 0.83 (0.03, 17.48) | 2.61 (0.94, 7.26) |
| ≥5 | 50 (43.10) | 66 (56.90) | 1.44 (0.01, 161.6) | 15.84 (4.04, 62.15) |
| < 2 (ref) | 239 (78.10) | 67 (21.90) | 1.00 | 1.00 |
| **History of abortion** | | | | |
| Yes | 60 (43.48) | 78 (56.52) | 2.13 (0.02, 20.16) | 3.67 (1.53, 8.83) |
| No | 506 (75.52) | 164 (24.48) | | 1.00 |
| **Maternal complications** | | | | |
| Yes | 86 (63.70) | 49 (36.30) | 1.13 (0.06, 18.74) | 1.35 (0.50, 3.62) |
| No (ref.) | 480 (71.32) | 193 (28.68) | 1.00 | 1.00 |
| **Sex of the newborn** | | | | |
| Male | 285 (65.22) | 152 (34.78) | 1.35 (0.02, 68.31) | **3.21 (1.33, 7.76)** |
| Female (ref.) | 281 75.74) | 90 (24.26) | 1.00 | 1.00 |
| **Birthweight (in grams)** | | | | |
| <1000 | 3 (22.00) | 22 (88.00) | 24.26 (1.11, 530.6) | 1.43 (0.29, 7.00) |
| <1000 to 1499 | 116 (50.22) | 115 (49.78) | 21.63 (5.25, 89.15) | 1.49 (0.67, 3.29) |
| 1500 to <2000 (ref.) | 447 (80.98) | 105 (19.02) | 1.00 | 1.00 |
| **Born preterm** | | | | |
| Yes | 462 (71.85) | 181 (28.15) | 3.05 (0.12, 56.93) | **8.56 (1.59, 46.14)** |
| No (ref.) | 104 (63.03) | 61 (36.97) | 1.00 | 1.00 |
| **Birth size-for-gestational-age** | | | | |
| Preterm-SGA | 208 (64.40) | 115 (35.60) | 2.23 (0.01, 36.84) | 0.97 (0.42, 2.27) |
| Term-SGA | 104 (63.03) | 61 (36.97) | 23.22 (0.014, 1.52) | 1.41 (0.41, 4.85) |
| LGA | 29 (85.29) | 5 (14.71) | 0.07 (0.004, 1.23) | 1.20 (0.07, 19.22) |
| AGA (ref.) | 225 (78.67) | 61 (21.33) | 1.00 | 1.00 |
| Neonatal complications (n = 592) | | | | |
| Yes | 236 (60.36) | 155 (39.64) | 345 (44.98, 264) | **4.68 (1.49, 14.76)** |
| No (ref.) | 184 (91.54) | 17 (8.46) | 1.00 | 1.00 |
| Wash immediately after birth (n = 517) | | | | |
| Yes | 59 (31.55) | 128 (68.45) | 88.64 (34.20, 229) | **20.85 (6.34, 68.51)** |
| No (ref.) | 296 (89.70) | 34 (10.30) | 1.00 | 1.00 |
| Hours after birth first put the baby on the breast (n = 790) | | | | |
| BF has not yet initiated | 89 (37.24) | 150 (62.76) | 36.87 (16.44, 82.7) | **7.40 (1.84, 29.74)** |
| BF initiated within 1 to < = 24 hrs | 211 (90.95) | 21 (9.05) | 0.58 (0.27, 1.21) | 1.95 (0.49, 7.71) |
| BF initiated after 24 hours | 78 (65.55) | 41 (34.45) | 10.93 (4.19, 28.52) | 2.91 (0.66, 12.75) |

*(Continued)*

**Table 6.** (Continued)

| Characteristics | Neonatal mortality | | CHR (95% CI) | AHR (95% CI) |
|---|---|---|---|---|
| | Censored (n = 566) n(%) | Died (n = 242) n(%) | | |
| Initiated at <1hr | 173 (86.50) | 27 (13.50) | 1.00 | |
| NICU admission | | | | |
| No | 269 (72.31) | 103 (27.69) | 0.49 (0.006, 39.67) | **3.10 (1.40, 6.90)** |
| Yes (ref.) | 297 (68.12) | 139 (31.88) | 1.00 | |
| Receiving effective KMC during KMC initiation (n = 780) | | | | |
| No | 223 (50.91) | 215 (49.09) | 530.74 (133, 2103) | **3.54 (1.14, 11.02)** |
| Yes (ref.) | 318 (92.98) | 24 (7.02) | 1.00 | |
| Receiving effective KMC during discharge (n = 611) | | | | |
| No | 94 (47.72) | 103 (52.28) | 13.19 (7.3, 24.75) | **9.57 (3.58, 25.61)** |
| Yes (ref.) | 352 (85.02) | 62 (14.98) | 1.00 | |
| ln(p) | P-value = 0.001 | | | 0.76 (0.46, 1.06) |
| P | | | | 2.15 (1.59, 2.90) |
| Theta (θ) | | | | 1.56 (0.74, 3.28) |

LR test of theta (θ) = 0: Chi square = 18.31 P-value = 0.001.

likely to die within a short time compared with those who had initiated BF (p-value = 0.001) (S4 Fig). The incidence rates of mortality were $52.65/10^3$, $5.99/10^3$, $4.02/10^3$, $20.39/10^3$ for neonates who had not yet initiated, initiated within <1 hr, initiated within 1 to <24 hrs, and initiated after 24 hrs of birth, respectively.

## Predictors of neonatal mortality: Bivariate analysis

To evaluate the application of mixed-effects survival (frailty) model, ln(-ln(S(t))) was plotted against ln(t). S5 Fig which depicts the plots of ln[-lnS(t)] against ln(t) for the categories of birth size-for-gestational-age. The curves in the graph seem to have nearly the same slope (i.e., are parallel, same p) signifying that the PH (and thus the AFT) assumptions hold. This signals the effect of exposure constantly rises over time and proposing the Weibull individual frailty regression assumptions is evenhanded. To select the best-fitting model (post estimation), the AIC and BIC were used. In this criteria, the preferred model is the one with the lowest value of the AIC or BIC. Consequently, the Weibull gamma frailty (univariate) model in the PH metric was the best well-fitted model (S2 Table). A statistical approach that integrates both distal and proximate determinants had been employed in our study. The final model incorporates both distal and proximal factors that were significant in the bivariate frailty regression model. In the bivariate analysis of socio-demographic characteristics (distal factors), only living children in the household were significant at a p-value of < 0.25 (S3 Table) and were considered in the multivariable individual frailty regression model.

In the bivariate analysis of proximal determinants (maternal and neonatal factors), gravidity, maternal age at birth, maternal complication, history of abortion, sex of the baby, birthweight, birth size-for-gestational-age, gestational age at birth, and neonatal complications were significant. In other bivariate analyses of proximal determinants (newborn care practices), hours after birth first put the infant into the breast, the timing of first bath, NICU admission, and effective KMC within 24 hours of birth and at discharge were significantly associated with time-to-death so that these variables were also eligible for the multivariable analysis. Mother's

occupation, father's age, monthly income, mother's marital status, mother's education, father's education, history of stillbirth, place of birth, and who assisted the delivery, were not significant in the bivariate analysis.

## Predictors of neonatal mortality: Multivariable analysis, the final model

In the model, the LR test for θ = 0 gives a statistically significant p-value of 0.01, signifying that the frailty element contributes to the model (clue towards the existence of undetected heterogeneity). This frailty term may denote the comprehensive effect of unmeasured variables on the survival of newborns. The estimate for the Weibull shape parameter (p) = 2.15 signifies increasing individual-level hazard over time, suggesting that the exponential model is not appropriate (Table 6). In this individual frailty regression model, the number of living children, history of abortion, sex of the newborn, gestational age at birth, neonatal complications, timing of the first bath after birth, hours after birth first put the baby into the breast, NICU admission and effective KMC during KMC initiation and discharge were found to be independent predictors of time-to-failure. These variables were adjusted for the mother's marital status, father's education, mother's age at birth, birth interval, and maternal complications.

When applying the individual frailty model, the reported AHRs carry the standard explanation only if relating two hazards conditional on given frailty (α) The estimated AHR for the covariate number of living children can be explained as "all other things equal (including the frailty α), neonates whose mothers had ≥ 5 living children had 16 times (AHR 15.84 95% CI 4.04, 62.15) shorter time to die than neonates of the mother who had less than two living children in the household (**Table 6**). LBW newborns born from mothers who had a history of abortion had higher hazard ratios of mortality compared with their counterparts (AHR 3.67 95% CI 1.53–8.83).

The hazard of mortality for male neonates was 3.21 times more likely compared with the hazard of female neonates (95% CI 1.33–7.76). Those babies who were born preterm had a higher rate of mortality compared with their term counterparts. Preterm neonates were 8.56 times more likely to die when compared with term neonates (95% CI 1.59–46.14). Neonates who had neonatal complications (like RDS, infections, and birth asphyxia) had a 4.68 (95% CI 1.49–14.76) times higher risk of death compared with those who haven't. Compared with babies who were not washed within 24 hours of birth, newborns who were washed immediately after birth (within 24 hours) had 20.85 times higher hazards of mortality (95% CI 6.34–68.51). Neonates who have not yet initiated breastfeeding had seven times (AHR 7.40 95% CI 1.84–29.74) more likely to die than those who initiated breastfeeding within one hour after birth. Those babies who were not admitted to NICU had 3 times (95% CI 1.40–6.90) shorter time-to-death compared with those who were admitted to NICU. Furthermore, not having effective KMC increases the risks of neonatal mortality 3.54 times compared with having effective KMC at the points of KMC initiation (95% CI 1.14–11.02) and 9.57 times immediately before discharge (95% CI 3.58–25.61 (Table 6).

## Discussion

This study was designed to determine NMRs and identify its independent predictors among LBW neonates admitted to KMC and/or NICU units of five selected hospitals in the area. Having too many children, having previous history of abortion, being born male and preterm, neonatal complications, immediate bathing of the baby, delay in commencement of breastfeeding, and not practicing effective KMC were independent predictors of time-to-death. According to this study, the overall incidence rate of mortality among babies born <2000g was 16.17/1000 neonatal-days of observation (95% CI 14.21, 18.32). Around 3% of the LBW babies were

ELBW whereas 29% and 68% of them were VLBW and LBW babies, respectively. This is comparable with the findings of earlier studies in the country [16,20]. On the contrary, it seems higher as compared to prior studies done in different parts of Ethiopia [25–27]; and African countries such as Cameroon [28] and Burkina Faso [29]. The mortality among ELBW babies was high (88%), while it was 49.78% among VLBW babies and 19.02% among LBW babies. This result was greater than other studies done, in India [30,31], Johannesburg [32], Northeast Brazil [33], and Mexico City [34]. The reported higher birth weight-specific NMRs in the current study could be partly due to complications of PTBs since the majority (79%) of the babies in this study were extremely and/or very preterm and they might have a difficult time eating, to get back their birthweight and fight infections. The inconsistency might also be allied with a true high incidence of mortality. Besides, low accessibility and poor quality [35] of neonatal care and poor feeding practices might increase the hazard of death among LBW neonates. The disagreement also could be ascribed to the disparity in study design and population.

The reported early and late neonatal mortality incidence rates were consistent with a study done in northwest Ethiopia [27], while it was greater than a study done in the eastern part of Ethiopia [36]. The inequality could be described, in part, to the dissimilarities in study participants, study settings, and sample size. Most of the neonatal deaths were in the first 24 hours (16.12%) and seven days (77.69%) of birth. This finding is consistent with earlier studies done in northwest Ethiopia [20] and in Butajira, Ethiopia [37]. This might be due to complications happening during the prenatal period [38,39]. This might also be clarified by Gizaw [37] a poorer setup/system of the health facilities where the birth happened. It might also be linked to delay in detection and inadequate warmth (only 47% of the newborns were got effective KMC since discharge) and lack of breastfeeding support (65% of the newborns given colostrum, and 66.01% breastfed in the last 24 hours since discharge). We noted that greater than half of the neonatal deaths (55.23%) occurred among newborn babies who were born both SGA and preterm. It is analogous to findings from East African countries [7] and India [40].

Among the distal factors, only mothers who had $\geq$ five children were statistically related to neonatal mortality. Among predictors classified under maternal characteristics having previous history of abortion was statistically significant. Mothers of newborns who had previous history of abortion had a greater hazard of mortality as compared to those who had not any history of abortion. This finding is consistent with a study done in Ethiopia [41]. This can be justified as having previous history of abortion is related to negative pregnancy outcomes like bleeding in the first trimester, PTB, LBW, and neonatal mortality [41,42] In this study, gender, preterm birth, neonatal complications and effective KMC were independent predictors of neonatal mortality. This finding is consistent with other previous studies that being male was related to a higher risk of mortality than being female [33,43–48]. This could be as Zeitlin J et al. [49] described, although the mean birth weight is greater in males than females, males are more probably to be born preterm. Furthermore, males had more postnatal complications, including RDS and sepsis [50,51].

Being born preterm was found to raise the likelihood of mortality. These are consistent with studies done in the country and abroad [26,29,33,41,44,46,52–56]. This could be explained as Zage [57] described, the smaller the birth weight and the gestation the greater the chance of having neonatal death due to lung immaturity and immune systems linked with ELBW and extremely PTBs. The greater peril of death among PTBs could be described as these babies perhaps having more neonatal infections and complications [58].

Newborns who washed instantaneously after birth (<24 hours) had greater hazards of mortality compared with babies who were not washed within one day of birth. Early bathing has shown a significant increase in the occurrence of hypothermia and has to be postponed to at least 6 hours after birth[59]. Newborns who were driven into the breast instantaneously after

birth had lower hazards of mortality than those who had yet not initiated breastfeeding (or late initiated). But this result needs to be cautiously interpreted. Since late commencement of breastfeeding may be causally related to severe sicknesses and vice versa[60]. Late commencement of breastfeeding is related to a greater threat of numerous neonatal complications[61]. Early commencement of breastfeeding might be important in reducing neonatal illness/infection related to neonatal mortality[62]. The higher neonatal deaths among newborns who had not initiated breastfeeding or initiated lately are partly due to neonatal illness/complications. This study is in line with other studies[27,43,44,56,63,64] that put the newborns into the breast instantly after birth to advance neonatal survival. As Lunze[65] described, LBW newborns are a greater threat of dropping body temperature as a result of immature thermal regulation. Breastfeeding supports warm-up[59].

In our study, practicing effective KMC lowers the hazards of deaths of LBW babies. This result is consistent with a study done in South Africa [32], not practicing KMC increased the hazards of mortality. A meta-analysis study's findings support this evidence, studies that involved effective KMC in their KMC intervention parcels showed a stronger preventive impact of KMC against death than studies using any other KMC definitions [66]. This might be as elucidated by the meta-analysis result findings [66]; KMC decreased the threat of sepsis, hypothermia, and hypoglycemia and improve EBF among LBW babies. KMC also lowers the threat of illness, and hospital infections, and improves maternal and child attachment [67,68].

The current study has certain limitations. One of the major limitations is that the high mortality rate implies an inadequate quality of care at these facilities, but the study has few measures of quality of care by providers and no measures of the facilities themselves. Secondly, this study did not include home deliveries and newborns who arrived at study hospitals after 3 days of life which may underestimate the percentage of mortality because home deliveries are at risk of complications and deaths. In addition, newborns who were referred outside the study hospital within three days of life were excluded, which may also underestimate the frequency of mortality. The gestational age calculation was mainly laid on the last menstrual period hence may introduce recall bias. Finally, the birth size-for gestational age reference charts we applied were the Intergrowth 21st, (international growth standard) which may probably overestimate the numbers of SGAs in the Ethiopian population.

## Conclusions

Overall, the study has revealed that the incidence rate of mortality among babies born <2000g was unacceptably high, given the overall Ethiopian 2019 neonatal population estimates. The ENMRs and LNMRs were very high. More than half of the deaths have occurred among LBW (<2000g) newborns who were born both preterm and SGA.

The factors that increase the threat of death among LBW neonates are amendable. These include: having too many children, preterm birth, neonatal complications, immediate bathing of the baby, delay in commencement of breastfeeding, and not practicing effective KMC. The higher hazards of death among the smallest newborns together with an improved drift of facility delivery reveal the requirement of additional investment in the scale-up of evidence-based interventions (like KMC in Ethiopia).

Unlimited effort must be given concerning the care of babies born <2000g with much emphasis on preterm and small babies to meet the sustainable development goal (SGD) three. In addition, extensive efforts to enhance the early commencement of breastfeeding and practicing effective KMC might help to lower the hazards of death in this population in Ethiopia. Additional nutritional supplementation is very essential during pregnancy to decrease SGA and then mortality.

## Supporting information

**S1 Fig. Conceptual framework in the study of survival status and predictors of neonatal mortality among LBW neonates in Oromia Region and Addis Ababa City, Ethiopia, 2019.** (DOCX)

**S2 Fig. The Kaplan-Meier failure estimates compare time-to-death of neonate with categories of birth size-for-gestational-age among babies born <2000g, Ethiopia, 2017–2019.** (DOCX)

**S3 Fig. The Kaplan-Meier failure estimates compare time-to-death of neonate with categories of birthweight among neonates admitted to KMC unit, Ethiopia, 2017–2019.** (DOCX)

**S4 Fig. Kaplan-Meier failure estimates compare time-to-death of LBW newborns with categories of hours after birth first put the baby to the breast among LBW neonates admitted to KMC units of five public hospitals in Oromia Region and Addis Ababa City, Ethiopia, 2017–2019.** (DOCX)

**S5 Fig. Graphical evaluation of the Weibull frailty regression model assumption by using the covariate birthweight-for-gestational-age for the study, Ethiopia, 2019.** (DOCX)

**S1 Table. Life table showing the cumulative probability of surviving stratified by birth size-for-gestational-age among LBW neonates admitted to KMC units of five Public Hospitals in Oromia region and Addis Ababa City, Ethiopia, 2017–2019.** (DOCX)

**S2 Table. The log likelihood of different parametric survival model along with their AIC and BIC.** (DOCX)

**S3 Table. Bivariate individual frailty regression model showing CHRs and p-values in the study of predictors of mortality among LBW neonates admitted to KMC units of five public hospitals in Oromia Region and Addis Ababa City, Ethiopia.** (DOCX)

**S1 Dataset. Complete data set.** (XLSX)

## Acknowledgments

We are grateful to the Arba Minch University, Addis Ababa University, and Harvard T.H. Chan School of Public Health and WHO for their technical support. We would also like to thank Emma Williams, all the study participants, data collectors, supervisors, and all KMC scale-up study teams (Oromia site).

## Author Contributions

**Conceptualization:** Mesfin Kote Debere, Damen Haile Mariam, Ahmed Ali, Amha Mekasha, Grace J. Chan.

**Data curation:** Mesfin Kote Debere, Grace J. Chan.

**Formal analysis:** Mesfin Kote Debere, Grace J. Chan.

**Funding acquisition:** Damen Haile Mariam, Grace J. Chan.

**Investigation:** Mesfin Kote Debere, Damen Haile Mariam, Ahmed Ali, Amha Mekasha, Grace J. Chan.

**Methodology:** Mesfin Kote Debere, Grace J. Chan.

**Project administration:** Mesfin Kote Debere, Damen Haile Mariam, Ahmed Ali, Amha Mekasha.

**Resources:** Damen Haile Mariam, Ahmed Ali, Amha Mekasha, Grace J. Chan.

**Software:** Mesfin Kote Debere.

**Supervision:** Damen Haile Mariam, Ahmed Ali, Amha Mekasha, Grace J. Chan.

**Validation:** Mesfin Kote Debere, Damen Haile Mariam, Ahmed Ali, Grace J. Chan.

**Visualization:** Amha Mekasha.

**Writing – original draft:** Mesfin Kote Debere.

**Writing – review & editing:** Damen Haile Mariam, Ahmed Ali, Amha Mekasha, Grace J. Chan.

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
