## [Decision Letter · Decision Letter 0]

25 Nov 2021

PONE-D-21-16170Survival status and predictors of mortality among low -birth-weight neonates admitted to KMC units of five public hospitals in Ethiopia: frailty survival regression modelPLOS ONE

Dear Dr. Debere,

Thank you for submitting your manuscript to PLOS ONE. After careful consideration, we feel that it has merit but does not fully meet PLOS ONE’s publication criteria as it currently stands. Therefore, we invite you to submit a revised version of the manuscript that addresses the points raised during the review process.

We look forward to receiving your revised manuscript.

Kind regards,

Funmilola M. OlaOlorun, PhD

Academic Editor

PLOS ONE

Journal Requirements:

2. Please provide additional details regarding participant consent. In the ethics statement in the Methods and online submission information, please ensure that you have specified whether consent was informed.

Additional Editor Comments (if provided):

This manuscript needs revisions to the Methods section in order to strengthen the quality. Reviewer 2 makes important observations and suggestions on how to improve the manuscript.

Reviewers' comments:

Reviewer's Responses to Questions

**Comments to the Author**

1. Is the manuscript technically sound, and do the data support the conclusions?

Reviewer #1: Yes

Reviewer #2: Yes

2. Has the statistical analysis been performed appropriately and rigorously? 

Reviewer #1: Yes

Reviewer #2: No

3. Have the authors made all data underlying the findings in their manuscript fully available?

Reviewer #1: Yes

Reviewer #2: Yes

4. Is the manuscript presented in an intelligible fashion and written in standard English?

Reviewer #1: Yes

Reviewer #2: Yes

5. Review Comments to the Author

Reviewer #1: Thanks for the authors for addressed my comments. They addressed my review comments properly and the revised version reads well. I have no further comment on the paper. I recommend that this manuscript can be accepted for publication.

Reviewer #2: Survival status and predictors of mortality among low -birth-weight neonates admitted to KMC units of five public hospitals in Ethiopia: frailty survival regression model

Manuscript Number: PONE-D-21-16170R

Comments to Authors and Editor

General comments

The authors attempt to determine the incidence of neonatal mortality rates and identify its

predictors among LBW neonates born or treated at hospitals using frailty regression model.

The study looks interesting; however, my main concern is the clarity of the presentation from methodological point of view. I think most readers would find it very difficult to implement the described approach in other data sets. The manuscript would be more strengthened by revising the methodology and results to make clear understanding.

Parametric frailty regression model

I couldn’t find anywhere in the manuscript how to address the problem of heterogeneity in a population resulting from unobserved covariates. I prefer more justification to use frailty models in this setting, then readers can apply same analytical framework to their study. I strongly recommend avoiding mathematical notation as there is no implication for these notations in the result discussion.

Model building

Model building is always a risky business. Adding significant number of exposures, confounders and predictors looks more challenging for appropriate model building. I strongly suggest Authors to go through the scientific process of variables selection (lasso) in the model building strategy.

Presentation

The manuscript is overly long with a lot of redundancies. In this circumstance, I strongly recommend making shorter with clear message to the audience about mortality among LBW new-borns.

6. PLOS authors have the option to publish the peer review history of their article (what does this mean?). If published, this will include your full peer review and any attached files.

Reviewer #1: **Yes: **Gizachew Tessema

Reviewer #2: **Yes: **Shahid Ullah

---

## [Author Response · Author response to Decision Letter 0]

8 Feb 2022

Manuscript Number: PONE-D-21-16170R 

Response to Reviewers

Dear Dr. Funmilola M. OlaOlorun,

Academic Editor, 

PLOS ONE, 

Thank you for giving us the opportunity to submit a revised draft of the manuscript “Survival status and predictors of mortality among low -birth-weight neonates admitted to KMC units of five public hospitals in Ethiopia: frailty survival regression model” for publication in the PLOS ONE Journal. We appreciate you and the reviewers for your precious time in reviewing our paper and providing valuable comments. It was your valuable and insightful comments that led to possible improvements in the current version. The authors have carefully considered the comments and tried our best to address every one of them. We hope the manuscript after careful revisions meet your high standards. The authors welcome further constructive comments if any. Below we provide the point-by-point responses. Please see, in blue, for a point-by-point response to the reviewers. All modifications in the manuscript have been highlighted in track changes.

Responses to points raised by editors

We tried to prepare the manuscript according to PLOS ONE's style requirements. If any, we are ready to revise it again.

2. Please provide additional details regarding participant consent. In the ethics statement in the Methods and online submission information, please ensure that you have specified whether consent was informed.

All respondents were informed about the study procedures and the interview was conducted after obtaining their informed consent. The participants were assured of the confidentiality as they participate in the investigation.

Accepted and corrected

4. In your Data Availability statement, you have not specified where the minimal data set underlying the results described in your manuscript can be found. PLOS defines a study's minimal data set as the underlying data used to reach the conclusions drawn in the manuscript and any additional data required to replicate the reported study findings in their entirety. All PLOS journals require that the minimal data set be made fully available.

The minimum data set is fully available within the manuscript.

Responds to the reviewer’s comments:

Thank you Dr. Shahid Ullah for your deep review of the paper

Reviewer #2:

Response to comment: The authors attempt to determine the incidence of neonatal mortality rates and identify its predictors among LBW neonates born or treated at hospitals using frailty regression model.

The study looks interesting; however, my main concern is the clarity of the presentation from methodological point of view. I think most readers would find it very difficult to implement the described approach in other data sets. The manuscript would be more strengthened by revising the methodology and results to make clear understanding.

Thank you for the suggestion. We have re-written this part according to the Reviewer’s comments. Considering the Reviewer’s suggestion, we add a detailed account of many of the methods parts of the manuscript.

Reviewer #2:

Response to comment: I could not find anywhere in the manuscript how to address the problem of heterogeneity in a population resulting from unobserved covariates. I prefer more justification to use frailty models in this setting, then readers can apply same analytical framework to their study. I strongly recommend avoiding mathematical notation, as there is no implication for these notations in the result discussion.

We are very sorry for not including the justification to use frailty model in this setting. Considering the Reviewer’s suggestion, we add a detailed account of the justification to use frailty model. In this study, neonates that were treated in the same hospital might share similar facilities, physicians, midwifes, and other neonatal care practices. They might also share some similarities that were not measured as covariates and varied by hospital. These hospital effect might create dependence between the study outcomes at each hospital and lead the failure times to correlate. Therefore, to account for this correlation between individuals within clusters (in this case hospitals), a random effects survival model, parametric shared frailty model, was employed to find out independent predictors of neonatal mortality. Here, frailty describes hospital-to-hospital variations that were not explained by observed covariates. Shared frailty model assumes that individuals in a subgroup (in the same hospital) share the same frailty (α), but frailty from group to group may differ. A one-parameter gamma frailty distribution with mean 1 and variance θ, and Weibull distribution for the baseline hazard, was employed.

Reviewer #2:

Response to comment: Model building is always a risky business. Adding significant number of exposures, confounders and predictors looks more challenging for appropriate model building. I strongly suggest Authors to go through the scientific process of variables selection (lasso) in the model building strategy.

Thank you for the suggestion. We tried to follow a scientific procedure, stepwise regression (backward method) for variable selection. As you know, most recommend the lasso method when there is high dimensional data (the number of variables are many, specifically if the number of independent variables are much greater than the sample size). We have few variables and a large sample size (n=808) in this manuscript. We have greater than thirty (30) respondents/or the number of cases for one variable. Twenty-five (25) variables were entered for the bivariate analysis. From these, only fifteen (15) were significant in the bivariate analysis and were entered for multivariable analysis. To prevent overfitting due to either collinearity of the covariates or high dimensionality, we did also a multicollinearity check through VIF and there were no variables that were correlated to each other. We hope the results will be nearly similar if done with other methods of variable selection. Variable selection for survival data analysis poses many challenges because of the complicated data structure. The lasso method predicts the coefficients, not the hazard ratios. Another challenge we pose is the lasso method is mainly works for linear, logit, probit, and Poisson models. I hope you will understand. If you have further concern please let me know before decision.

Reviewer #2:

Response to comment: The manuscript is overly long with many redundancies. In this circumstance, I strongly recommend making shorter with clear message to the audience about mortality among LBW newborns.

Thank you for the suggestion. We have tried to avoid the redundancies and re-written the manuscript according to the Reviewer’s comments

We tried our best to improve the manuscript and made some changes in the manuscript. We appreciate for Editors/Reviewers’ warm work earnestly, and hope that the correction will meet with approval. 

Once again, thank you very much for your comments and suggestions.

---

## [Decision Letter · Decision Letter 1]

20 Jul 2022

PONE-D-21-16170R1Survival status and predictors of mortality among low -birth-weight neonates admitted to KMC units of five public hospitals in Ethiopia: frailty survival regression modelPLOS ONE

Dear Dr. Debere,

Thank you for submitting your manuscript to PLOS ONE. After careful consideration, we feel that it has merit but does not fully meet PLOS ONE’s publication criteria as it currently stands. Therefore, we invite you to submit a revised version of the manuscript that addresses the points raised during the review process.

We look forward to receiving your revised manuscript.

Kind regards,

Funmilola M. OlaOlorun, PhD

Academic Editor

PLOS ONE

Additional Editor Comments (if provided):

The reviewers of the revised version of this manuscript have expressed concerns about your choice of a frailty model, and the reproducibility of the methods of this research. If you choose to revise further, please ensure you are intentional about describing and justifying your methodological approach. In addition, more can be done to improve the clarity and readability of the manuscript.

Reviewers' comments:

Reviewer's Responses to Questions

**Comments to the Author**

1. If the authors have adequately addressed your comments raised in a previous round of review and you feel that this manuscript is now acceptable for publication, you may indicate that here to bypass the “Comments to the Author” section, enter your conflict of interest statement in the “Confidential to Editor” section, and submit your "Accept" recommendation.

Reviewer #3: (No Response)

Reviewer #4: (No Response)

2. Is the manuscript technically sound, and do the data support the conclusions?

Reviewer #3: Yes

Reviewer #4: Yes

3. Has the statistical analysis been performed appropriately and rigorously? 

Reviewer #3: Yes

Reviewer #4: Yes

4. Have the authors made all data underlying the findings in their manuscript fully available?

Reviewer #3: No

Reviewer #4: No

5. Is the manuscript presented in an intelligible fashion and written in standard English?

Reviewer #3: Yes

Reviewer #4: No

6. Review Comments to the Author

Reviewer #3: Comments

Introduction

The rational of the research is not clear and not novel. Many studies were conducted on this area that accounted multiple predictors. What makes this study unique?

Methods

It is not clear how the sample size is determined. How do you know the sample size used is sufficient enough for the analysis method you employed?

Sampling method; what do you mean by consecutive? Do you mean you included everyone during the study period? Is that a kind of cluster sampling?

You do not need “variable of the study” session

Outcome variable definition seems binary. But for this research it is time to event. So, correct your outcome variable definition.

Data collection instrument and process: How did you track those who were discharged before 28 days? How frequent you visited the new-born after discharge?

How could you treated twins and triplets as independent? I suggest you to take one among the twin or triplets.

Data analysis: the choice of the statistical model (random effect survival) seems strange. Since the number of clusters (hospitals) are very few frailty may not be appropriate.

Results

The descriptive results were not presented based survival data analysis form. Where is information related to the life table? Table 1-3 should be changed as per the comment.

Person time of observation (the time a risk) has to be the first information presented in this part.

All the percentages should take in to account person time of observation.

The survival curve is not correctly presented (Figure 3). This should be modified and replaced with the standard presentation of the survival curve.

Table 5 should be modified and the number of event for each category of the variables should be presented.

Discussion

The first paragraph should present all the necessary findings of the paper.

Reviewer #4: This is important work, specially for Ethiopia, and the authors have made great efforts to address comments in a prior review. While the conclusions are not really new, this may be important in the national policy setting context. A few issues remain:

1. Babies transferred to an outside facility after 3 days of life were excluded. Since there is a sizable number of babies fitting this criterion, and no information is provided on the characteristics of these excluded babies vs those in the analysis, it is unclear what effect this has on the results and conclusions.

2. The sample size calculation approach is poorly explained. Is it based on the precision/width of the confidence interval for an assumed rate of NMR? Please clarify.

3. The model selection approach may not be optimal. First, basing covariate selection strictly on empirical bivariate analysis results by definition cannot identify hidden confounders and higher level interactions. Second, using AIC is more appropriate for developing a prediction model and not necessarily for identifying causal pathways or markers for the outcome.

4. Presentation of stratified/subgroup analysis should be ideally preceded by and conditional on a significant interaction test (between SGA and prematurity in this instance).

5. It is unclear if CHR (table 5) is ever defined.

6. The authors state that all the relevant data is available within the manuscript, but I think the journal is asking for individual and not aggregate level data. I did not see the former included.

7. PLOS authors have the option to publish the peer review history of their article (what does this mean?). If published, this will include your full peer review and any attached files.

Reviewer #3: No

Reviewer #4: No

---

## [Author Response · Author response to Decision Letter 1]

12 Aug 2022

Manuscript Number: PONE-D-21-16170R 

Response to Reviewers

Dear Dr. Funmilola M. OlaOlorun,

Academic Editor, 

PLOS ONE, 

Thank you for allowing us the opportunity to submit a revised draft of the manuscript “Survival status and predictors of mortality among low -birth-weight neonates admitted to KMC units of five public hospitals in Ethiopia: frailty survival regression model” for publication in the PLOS ONE Journal. We appreciate the time and effort that the reviewers have dedicated to providing valuable feedback on our manuscript, and are grateful for their constructive criticism. It was your valuable and insightful comments that led to possible improvements in the current version. We have now revised the manuscript to reflect most of the suggestions provided by the reviewers, and feel that it is significantly improved. We hope the manuscript after careful revisions meet your high standards. The authors welcome further constructive comments if any. Below we provide the point-by-point responses. Please see, in Blue, for a point-by-point response to the reviewers. All modifications in the manuscript have been highlighted in track changes.

Responses to points raised by editors

 The reviewers of the revised version of this manuscript have expressed concerns about your choice of a frailty model, and the reproducibility of the methods of this research. If you choose to revise further, please ensure you are intentional about describing and justifying your methodological approach. In addition, more can be done to improve the clarity and readability of the manuscript

Thank you for your suggestion. The notion of frailty provides a convenient way to introduce random effects, association, and unobserved heterogeneity into models for survival data. In its simplest form, frailty is an unobserved random factor that modifies multiplicatively the hazard function of an individual or a group or cluster of individuals. The goal is to estimate a “frailty” parameter that “absorbs” the effects of these unmeasured risk factors/covariates. In statistical terms, a frailty model is a random effect model for time-to-event data, where the random effect (the frailty) has a multiplicative effect on the baseline hazard function. It extend Cox proportional hazards (PH) model by introducing unobserved “frailties” to the model. 

In our study, there are unobserved covariates (e.g. nutritional status of the mother during pregnancy, weight gain during pregnacy) or unobservable heterogeneity, caused by different sources. We considered a random effects survival (frailty) model to take into account these unobserved effects. Initially, we planned to fit the shared frailty model to account for cluster effects or variations between clusters (hospitals) and to incorporate correlation within the same clusters since the same cluster shares the same random effects. We treat hospitals as clusters. Due to very few clusters, shared (clustered survival analysis) may not be appropriate, as the reviewers also suggested. We checked the best-fitting model using the information criterion (AIC and BIC). Based on these criteria, the univariate/ non-shared/ or individual frailty model is the best-fitting model among others. Frailty models for univariate data have been used to account for heterogeneous times-to-failure among individuals. Usually, it is thought of as unmeasured risk factors. If there are unmeasured or unobserved “frailties,” the hazard rate will not only be a function of the covariates, but also a function of the frailties. A frailty model in the univariate case introduces an unobservable multiplicative effect α on the hazard, so that conditional on the frailty: h(t|α) = αh(t). 

h(ti|αi) = αih(ti) = αih0(ti) exp(Xti β) ……… (1)

Where α is some random positive quantity assumed to have mean one and variance θ. Those individuals who possess α > 1 are said to be more frail for reasons left unexplained by the covariates and will have an increased risk of failure. Conversely, those individuals with α < 1 are less frail and will tend to survive longer all else being equal (i.e., given a certain covariate pattern). Since α is a multiplicative effect, it is easy to see from (1) how one can think of frailty as representing the cumulative effect of one or more omitted covariates. If α was not significant in this study, the model can be lowered into a fixed effects model whether parametric (e.g. Weibull regression) or semi-parametric (e.g. Cox), or others. Since the LR test of theta (θ) = 0 is significant ( P-value = 0.001), suggesting that the frailty element contributes to the model (clue towards the existence of undetected heterogeneity). Hence, the individual frailty regression model is the appropriate model for this study. The manuscript is also revised by considering this change.

Responds to the reviewer’s comments:

Thank you Reviewer #3 for your deep review of the paper

Reviewer #3:

Response to comment: The rational of the research is not clear and not novel. Many studies were conducted on this area that accounted multiple predictors. What makes this study unique?

First, thank you for the question. 

This study is conducted among low birthweight (LBW) babies (born <2000 grams). Many previous studies in Ethiopia report the mortality rates among LBW babies (not stratified rate). The main causes of LBW are preterm birth and growth restriction (SGA). Although preterm birth and SGA have common causes they have also different causes. Merging and reporting the mortality rate as LBW might hinder prevention mechanisms. So reporting the stratified incidence rates of mortality may advance the prevention mechanisms of LBW. In Ethiopia, if a baby born <1000 grams and/or <28 weeks of gestation is treated as abortion irrespective of viability. In this study as part of a large KMC implementation research project, we admitted all these newborns, if viable, to the KMC unit. Many babies who were <1000 grams and even at 24 weeks of gestation survived. In this study, we also evaluated effective KMC can advance the survival of very small babies in Ethiopia. This has an operational policy impact on the Ethiopian population.

Reviewer #3:

Methods sections:

Response to comment: It is not clear how the sample size is determined. How do you know the sample size used is sufficient enough for the analysis method you employed?

Now, the sample size calculated to estimate the incidence rate of mortality was checked whether it is enough or not to identify predictors of mortality. Using a two-sample comparison of survivor functions (Log-rank test, Freedman method), the number of events (E) that could be needed was calculated by: E = ((Zα/2 + Zβ)^2)/((ln(HR))^2 pq) in STATA software. Using Freedman principles q = rate of an event is equal to p = survival probability rate (proportional allocations, p = q, the median survival = 50%). To calculate the total number of events (E) that could be needed to compare the two groups (exposed and unexposed/1:1 ratio), we take α = 0.05 and β = 0.20. With these values of α and β, Zα/2 = 1.96 and Zβ = 0.84, and taking the HR 1.65 from previous study, and 10% withdrawal, the number of events (E) becomes 132. Then, the total number of participants needed (n) for the survival study was calculated by using n = E/Pr(event), and it becomes 407 by considering 1.5 clustering effect. We now mention these considerations in the methods section.

Reviewer #3:

Response to comment: Sampling method; what do you mean by consecutive? Do you mean you included everyone during the study period? Is that a kind of cluster sampling?

Thank you for the question. Yes, it is to mean everyone during the study period was included (i.e. cluster sampling). We now mention these considerations in the methods section.

You do not need “variable of the study” session

Outcome variable definition seems binary. But for this research it is time to event. So, correct your outcome variable definition.

Thank you for your suggestion. All the given suggestions are accepted and the manuscript is revised according to the reviewer’s comments.

Reviewer #3:

Response to comment: Data collection instrument and process: How did you track those who were discharged before 28 days? How frequent you visited the new-born after discharge?

Thank you for your suggestion. There were follow-up home visits at seven days of life, seven days post-discharge, and 29 days of life to track information if the baby was discharged before 28 days of life. We now mention these considerations in the methods section (data collection instrument and process).

Reviewer #3:

Response to comment:

How could you treated twins and triplets as independent? I suggest you to take one among the twin or triplets.

Thank you for the question. It can be done as you suggested. But, from a clinical point of view and from many other previous studies, the risk of mortality is increased with the number of births (singleton vs multiple births). Taking singleton and one from twins or triplets may underestimate the outcome of interest (mortality). So it is better to take all types of births to estimate the true incidence rate of mortality.

Reviewer #3:

Data analysis: the choice of the statistical model (random effect survival) seems strange. Since the number of clusters (hospitals) are very few frailty may not be appropriate.

Thank you for your suggestion. As you rightly said the number of clusters are very few (five), and the estimate may be unstable in case of shared frailty. We now fit and compare around nine (9) parametric survival models including the Weibull regression model and shared frailty regression model to see the well-fitted model for the data. Using the AIC and BIC, the individual/univariate frailty regression model (unshared) is the best-fitting model. Look S2 Table (in the supplementary materials). We now mention these considerations in the results section.

Reviewer #3:

Results

The descriptive results were not presented based survival data analysis form. Where is information related to the life table? Table 1-3 should be changed as per the comment.

Person time of observation (the time a risk) has to be the first information presented in this part. All the percentages should take in to account person time of observation. 

Thank you for your suggestion. All the comments are accepted and incorporated into the document. Life table showing the overall cumulative probability of surviving (Table 5) and birth size-for-gestational-age specific cumulative probability of surviving (S1 Table) were produced. The total time at risk is placed as you suggested and Tables 1-3 are also modified as you suggested (neonatal days of observation were taken into account to compute the incidence rate of mortality).

Reviewer #3:

The survival curve is not correctly presented (Figure 3). This should be modified and replaced with the standard presentation of the survival curve.

Thank you for your suggestion. The KM survival estimate is modified and replaced with the standard presentation of the survival curve (look at Fig. 3 and S2 Fig.)

Reviewer #3:

Table 5 should be modified and the number of event for each category of the variables should be presented.

Thank you for your suggestion. Revised as you suggested, the number of events and censured (with %) are now included. We now mention these considerations in Table 7.

Reviewer #3:

Discussion

The first paragraph should present all the necessary findings of the paper.

Thank you for your suggestion. We now mention these considerations in the discussion section.

Thank you Reviewer #4 for your deep review of the paper

Reviewer #4: 

This is important work, especially for Ethiopia, and the authors have made great efforts to address comments in a prior review. While the conclusions are not really new, this may be important in the national policy setting context. A few issues remain: 

1. Babies transferred to an outside facility after 3 days of life were excluded. Since there is a sizable number of babies fitting this criterion, and no information is provided on the characteristics of these excluded babies vs those in the analysis, it is unclear what effect this has on the results and conclusions.

Thank you for your suggestion. We checked the differences for baseline characteristics of babies included and excluded (like transferred to an outside facility after 3 days of life). According to the statistical test made, there were no statistically significant differences between babies who were included and excluded in this analysis. We now mention these considerations in the results section.

Reviewer #4: 

2. The sample size calculation approach is poorly explained. Is it based on the precision/width of the confidence interval for an assumed rate of NMR? Please clarify.

Thank you for your suggestion. 

However, we included everyone during the study period (n = 808), and the minimum required sample size was calculated using one group dichotomous outcome variable (mortality: died, censured) using STATA statistical package using the following formula:

n=((Zα√(po (1-Po) ) +Zβ√(P1 (1-P1)) ^^2)/(P1-Po)^2 

Assumptions used for the calculation were: neonatal mortality rate of 28% based on previous estimates (Po); we want to sample a group large enough to detect an incidence rate of 34% (i.e. H1: P1 = 0.34) if that is the real, true incidence. We set the two-sided α at 0.05 and want a power of 0.80 to detect the difference in incidence setting the rate of deaths of neonates in the area as high (P1 = 34%) to make the sample size large enough to detect the true incidence. And 10% lost-to-follow-up; and 1.5 for clustering effect. The sample size was also determined by considering the model that was fitted. We now mention these considerations in the methods section.

Reviewer #4: 

3. The model selection approach may not be optimal. First, basing covariate selection strictly on empirical bivariate analysis results by definition cannot identify hidden confounders and higher level interactions. Second, using AIC is more appropriate for developing a prediction model and not necessarily for identifying causal pathways or markers for the outcome.

Thank you for your suggestion. To evaluate the application of mixed-effects survival (frailty) model, ln(-ln(S(t))) was plotted against ln(t). The curves in the graph seem to have nearly the same slope (i.e., are parallel, have the same p) signifying that the PH (and thus the AFT) assumptions hold. This signals the effect of exposure constantly rises over time and proposing the Weibull individual frailty regression assumptions is evenhanded. To select the best-fitting model (post estimation), the AIC and BIC were used. In this criteria, the preferred model is the one with the lowest value of the AIC or BIC. Consequently, the Weibull gamma frailty (univariate) model in the PH metric was the best well-fitted model (S2 Table). We started with the classical bivariate analysis and then stepwise regression, for choosing predictor variables from a large set. Now, we considered AIC for model selection/post estimation (prediction model), not for covariate selection as you suggested. We now mention these considerations in the methods section.

Reviewer #4: 

4. Presentation of stratified/subgroup analysis should be ideally preceded by and conditional on a significant interaction test (between SGA and prematurity in this instance).

Thank you for your suggestion. Yes, the comment is accepted and the manuscript is revised as you suggested (Table 7 also includes the subgroup analysis).

Reviewer #4: 

5. It is unclear if CHR (table 5) is ever defined.

Sorry, it is to mean crude hazard ratios (CHRs) 

Reviewer #4: 

6. The authors state that all the relevant data is available within the manuscript, but I think the journal is asking for individual and not aggregate level data. I did not see the former included.

Thank you for your suggestion. Now all the relevant required data are within the manuscript. If the journal needs the database, we can upload it or can be accessed it upon request. 

We tried our best to improve the manuscript and made many changes in the manuscript. We appreciate for Editors/Reviewers’ warm work earnestly and hope that the correction will meet with approval. 

Once again, thank you very much for your comments and suggestions.

---

## [Decision Letter · Decision Letter 2]

7 Sep 2022

PONE-D-21-16170R2Survival status and predictors of mortality among low -birth-weight neonates admitted to KMC units of five public hospitals in Ethiopia: frailty survival regression modelPLOS ONE

Dear Dr. Debere,

Thank you for submitting your manuscript to PLOS ONE. After careful consideration, we feel that it has merit but does not fully meet PLOS ONE’s publication criteria as it currently stands. Therefore, we invite you to submit a revised version of the manuscript that addresses the points raised during the review process.

We look forward to receiving your revised manuscript.

Kind regards,

Funmilola M. OlaOlorun, PhD

Academic Editor

PLOS ONE

Journal Requirements:

Additional Editor Comments :

We thank the authors for their efforts in responding to the comments raised by the expert reviewers. Reviewer 3 would like to see additional minor edits before we can consider this manuscript for publication. Additionally, it would be great if the authors can read through again to edit grammatical and typographical errors since the journal does not offer any copy editing. Thank you.

Reviewers' comments:

Reviewer's Responses to Questions

**Comments to the Author**

1. If the authors have adequately addressed your comments raised in a previous round of review and you feel that this manuscript is now acceptable for publication, you may indicate that here to bypass the “Comments to the Author” section, enter your conflict of interest statement in the “Confidential to Editor” section, and submit your "Accept" recommendation.

Reviewer #3: (No Response)

Reviewer #4: All comments have been addressed

2. Is the manuscript technically sound, and do the data support the conclusions?

Reviewer #3: Yes

Reviewer #4: Yes

3. Has the statistical analysis been performed appropriately and rigorously? 

Reviewer #3: Yes

Reviewer #4: Yes

4. Have the authors made all data underlying the findings in their manuscript fully available?

Reviewer #3: No

Reviewer #4: No

5. Is the manuscript presented in an intelligible fashion and written in standard English?

Reviewer #3: Yes

Reviewer #4: Yes

6. Review Comments to the Author

Reviewer #3: The authors attempted to address the comments. However, I still have the following concerns;

1. The samplse size you defined (formula) should match the methods of analysis you used. Smple size for surviavl analysis is different from the sample size for proportion. Remove the formula and keep the text explanation.

2. Table 5 is not important to be presented in the document. KM can provide similar information instead of life table. So, remove table 5 from the document.

Reviewer #4: All comments this reviewer made in the previous review have been addressed. The manuscript still needs some editorial support for linguistic clarity and readability.

7. PLOS authors have the option to publish the peer review history of their article (what does this mean?). If published, this will include your full peer review and any attached files.

Reviewer #3: **Yes: **Tadesse Awoke Ayele

Reviewer #4: No

---

## [Author Response · Author response to Decision Letter 2]

2 Oct 2022

Manuscript Number: PONE-D-21-16170R 

Response to Reviewers

Dear Dr. Funmilola M. OlaOlorun,

Academic Editor, 

PLOS ONE, 

Thank you for allowing us the opportunity to submit a revised draft of the manuscript “Survival status and predictors of mortality among low -birth-weight neonates admitted to KMC units of five public hospitals in Ethiopia: frailty survival regression model” for publication in the PLOS ONE Journal. We appreciate the time and effort that the reviewers have dedicated to providing valuable feedback on our manuscript, and are grateful for their constructive criticism. It was your valuable and insightful comments that led to possible improvements in the current version. We have now revised the manuscript to reflect most of the suggestions provided by the reviewers, and feel that it is significantly improved. We hope the manuscript after careful revisions meet your high standards. The authors welcome further constructive comments if any. Below we provide the point-by-point responses. Please see, in Blue, for a point-by-point response to the reviewers. All modifications in the manuscript have been highlighted in track changes.

Responses to points raised by editors

Thank you for your suggestion. The reference list is checked, and it is complete and correct.

2. We thank the authors for their efforts in responding to the comments raised by the expert reviewers. Reviewer 3 would like to see additional minor edits before we can consider this manuscript for publication. Additionally, it would be great if the authors can read through again to edit grammatical and typographical errors since the journal does not offer any copy editing. Thank you.

Thank you for your suggestion. We made some grammatical and typographical edits.

Responds to the reviewer’s comments:

Thank you again both Reviewers

Reviewer #3 and #4

Response to comment: Have the authors made all data underlying the findings in their manuscript fully available?

We now uploaded the data set in Excell format. 

Reviewer #3:

Methods sections:

Response to comment: The authors attempted to address the comments. However, I still have the following concerns;

1. The samplse size you defined (formula) should match the methods of analysis you used. Sample size for surviavl analysis is different from the sample size for proportion. Remove the formula and keep the text explanation.

Thank you for your suggestion. The sample size calculation formula is deleted. We now mention these considerations in the methods section

2. Table 5 is not important to be presented in the document. KM can provide similar information instead of life table. So, remove table 5 from the document.

Thank you for your suggestion. Table 5 is removed as you suggested. We now mention these considerations in the results section

We tried our best to improve the manuscript and made many changes in the manuscript. We appreciate for Editors/Reviewers’ warm work earnestly and hope that the correction will meet with approval. 

Once again, thank you very much for your comments and suggestions.

---

## [Editor Report · Decision Letter 3]

5 Oct 2022

Survival status and predictors of mortality among low -birth-weight neonates admitted to KMC units of five public hospitals in Ethiopia: frailty survival regression model

PONE-D-21-16170R3

Dear Dr. Debere,

We’re pleased to inform you that your manuscript has been judged scientifically suitable for publication and will be formally accepted for publication once it meets all outstanding technical requirements.

Kind regards,

Funmilola M. OlaOlorun, PhD

Academic Editor

PLOS ONE
---

## [Editor Report · Acceptance letter]

12 Oct 2022

PONE-D-21-16170R3 

Survival status and predictors of mortality among low-birthweight neonates admitted to KMC units of five public hospitals in Ethiopia: frailty survival regression model 

Dear Dr. Debere:

I'm pleased to inform you that your manuscript has been deemed suitable for publication in PLOS ONE. Congratulations! Your manuscript is now with our production department. 

Kind regards, 

on behalf of

Dr. Funmilola M. OlaOlorun 

Academic Editor

PLOS ONE